# Omics Analysis for Dinoflagellates Biology Research

**DOI:** 10.3390/microorganisms7090288

**Published:** 2019-08-23

**Authors:** Yali Bi, Fangzhong Wang, Weiwen Zhang

**Affiliations:** 1Laboratory of Synthetic Microbiology, School of Chemical Engineering & Technology, Tianjin University, Tianjin 300072, China; 2Frontier Science Center for Synthetic Biology and Key Laboratory of Systems Bioengineering, Ministry of Education of China, Tianjin 300072, China; 3Collaborative Innovation Center of Chemical Science and Engineering, Tianjin 300072, China; 4Center for Biosafety Research and Strategy, Tianjin University, Tianjin 300072, China

**Keywords:** dinoflagellates, genomics, transcriptomics, proteomics, metabolomics, harmful algal blooms, toxin, symbiosis, lipid biosynthesis

## Abstract

Dinoflagellates are important primary producers for marine ecosystems and are also responsible for certain essential components in human foods. However, they are also notorious for their ability to form harmful algal blooms, and cause shellfish poisoning. Although much work has been devoted to dinoflagellates in recent decades, our understanding of them at a molecular level is still limited owing to some of their challenging biological properties, such as large genome size, permanently condensed liquid-crystalline chromosomes, and the 10-fold lower ratio of protein to DNA than other eukaryotic species. In recent years, omics technologies, such as genomics, transcriptomics, proteomics, and metabolomics, have been applied to the study of marine dinoflagellates and have uncovered many new physiological and metabolic characteristics of dinoflagellates. In this article, we review recent application of omics technologies in revealing some of the unusual features of dinoflagellate genomes and molecular mechanisms relevant to their biology, including the mechanism of harmful algal bloom formations, toxin biosynthesis, symbiosis, lipid biosynthesis, as well as species identification and evolution. We also discuss the challenges and provide prospective further study directions and applications of dinoflagellates.

## 1. Introduction

Dinoflagellates are unicellular protists that have two distinctive flagella during their life cycle [1]. The size of dinoflagellates ranges from a few micrometers to two millimeters [2]. The life styles of dinoflagellates include free living, parasitic, or mutualist [2]. So far nearly 4500 species of dinoflagellates have been isolated, and the majority of them (approximate 4000 species) are free living, while only a small percentage of them are found with parasitic or mutualist symbiotic life styles, such as *Syndinophyceae* [2]. Three major nutrient mechanisms, phototrophy, heterotrophy and mixotrophy, are all found and adopted by various dinoflagellates species, and among them approximately 50% of dinoflagellates are phototrophic [2,3].

Dinoflagellates receive great attention mostly due to their important roles in natural ecosystems, as well as being the producers of key components in human food (e.g., docosahexaenoic acids (DHA) as essential supplements) [4]. First, together with diatoms and coccolithophores, the photosynthetic species of dinoflagellates are among the most prominent primary producers in marine environments and play vital roles in the global carbon cycle [5]. Second, a number of photosynthetic dinoflagellates form a mutualism with other living organisms for maintaining the stability of ecosystems [6]. For example, reef-building corals are maintained based on symbiosis between the coral animal and *Symbiodiniaceae*, a photosynthetic dinoflagellate. Corals provide shelter and inorganic nutrients for *Symbiodiniaceae*, and in return, *Symbiodiniaceae* provides much more than energy metabolites to corals. They translocate antioxidants, signaling compounds, and are vitally important in nutrient cycling [6,7]. Finally, some heterotrophic dinoflagellates are important producers of highly-valued and essential food components. For example, DHA which has various beneficial effects on human health [8,9,10], such as improving cognitive development in infants and inhibiting hypertension, inflammation and certain cancers, cannot be synthesized by human beings, but accumulates in heterotrophic dinoflagellates *Crypthecodinium cohnii* [4]. The fraction of DHA in *C. cohnii* can be up to 25% of dry cell weight [11,12]. Therefore, the fermentation of *C. cohnii* has been considered as a sustainable and alternative way of producing high-quality DHA [4,12,13,14,15]. Meanwhile, dinoflagellates can also result in negative effects to the environment. One notorious phenomenon is “harmful algal blooms (HABs)”, also known as “red tides” [16], which are formed due to a sharp increase in dinoflagellates and bacteria, which can reach up to twenty million cells per liter along the coasts [2]. These blooms are often associated with the production of various secondary metabolites, which can be extremely toxic and can be more potent than some agents used in chemical warfare [17].

Dinoflagellates have additional prominent biological features that are worth of further exploration. For example, some species of dinoflagellates are capable of bioluminescence [18]. The bioluminescence systems consist of luciferase (enzyme), luciferin (substrate), and a protein that binds luciferin [19]. Bioluminescence seems to be separated by discrete particles called ‘‘scintillons’’ within cells [20]. One of the suggested functions is that it reduces the grazing behavior of copepods [21]. Some dinoflagellates have photosensitive organelles called “eyespots”, which consist of lipid droplets wrapped within stacked layers of membranes [22], and allow dinoflagellates to perform phototaxis: i.e., to move relative to the direction and intensity of light. In addition, dinoflagellate plastids originate from either secondary or tertiary endosymbiotic events, making them different from those found in plants or green alga [23]. The most cytologically prominent feature of dinoflagellates is their abnormal genome size and organization [24]. Typical dinoflagellate DNA content is estimated to be 3 to 250 pg·cell^−1^, equivalent to about 3000–245,000 Mb [25,26], roughly 1- to 80-fold larger than a human haploid genome [27]. Moreover, dinoflagellate nuclear DNA is widely methylated, of which 12–70% of thymine is replaced by 5-hydroxymethyluracil and cytosine is methylated to varying degrees [27,28,29]. In most eukaryotic organisms, these modifications are due to oxidative damage of thymine or 5-methylcytosine, which is quickly repaired by a DNA glycosylase [30]. Furthermore, the chromosomes of most dinoflagellates are permanently condensed without the aid of nucleosomes throughout some or all stages of their life cycle [24,31]. The protein content in the chromatin is usually low, with a ten-times lower protein: DNA ratio than that in other eukaryotes [16]. Dinoflagellates are the only known eukaryotes to apparently lack histone proteins [32]. Dinoflagellates are capable of recruiting other proteins, such as histone-like proteins from bacteria and dinoflagellate/viral nucleoproteins from viruses, as histone substitutes [32,33,34]. Finally, plastid genomes of dinoflagellates appear in the form of plasmid-like minicircles [35], and their mitochondrial genomes contain only three protein-coding genes and lack stop codons [36].

Due to their ecological and economical roles and nutrition contribution, a better understanding of dinoflagellate biology at the molecular level is important. In recent years, omics technologies [37], such as genomics, transcriptomics, proteomics, and metabolomics, have been successfully applied to the systematic analysis of dinoflagellates, in revealing the organization and evolution of dinoflagellates genomes [7,38,39,40,41], and molecular mechanisms related to HAB formation and toxin biosynthesis [42,43,44,45,46,47,48,49,50,51,52,53,54,55,56,57,58,59,60,61], symbiosis [39,40,41,62,63,64,65,66,67,68,69,70,71], and lipid accumulation [11,12,13,15,72,73,74,75,76] (Figure 1). In this review, we summarize recent progress on the omics studies of dinoflagellates biology. We also discuss the challenges and provide prospects for further study.

## 2. Omics Study on Dinoflagellates

### 2.1. Genomic Analysis

Genomic analyses help to examine the primary sequence and structural assembly of the complete genome of an organism [77]. Due to the high sequencing cost of traditional methods, referred to as Sanger sequencing, and the large sizes of dinoflagellates genomes, complete sequencing of a large number of nuclear genomes of dinoflagellates has long been out of reach [24]. With the advent and rapid development of next-generation sequencing (NGS), such as 454 sequencing, Solexa technology, the SOLiD platform, the Polonator, and the HeliScope Single Molecule Sequencer technology, the total cost of sequencing is sharply decreased and sequencing throughput is significantly enlarged [78,79,80], which has facilitated rapid, economic sequencing of a number of dinoflagellates species, with improved reliability and accuracy [7,38,39,40,41].

The most frequently used sequencing technologies for dinoflagellates are NGS 454 sequencing and Illumina Hiseq 2000/2500 (Table 1), which use clonal amplification and sequencing by synthesis to allow parallel sequencing [78]. Except for several thousand times higher sequencing throughput and the dramatically increased degree of parallelism, a major advantage of NGS over Sanger sequencing is the cost reduction by over two orders of magnitude [78]. Because the genomes of symbiotic species are among the smallest (1~5 Gbp) compared with other dinoflagellates [81], the first available draft genome of a dinoflagellate is *Breviolim minutum* (clade B) [38], soon followed by *Fugacium kawagutii* (clade F) [39], *Symbiodinium microadriaticum* (clade A) [40], *Cladocopium goreaui* (clade C) [7], and other *Symbiodiniaceae* species [41] (Table 1).

Detailed genomic analysis shows that the gene structures of *B*. *minutum* and *F. kawagutii* have several distinctive and divergent characteristics when compared to those of other eukaryotes. For example, analysis of genome sequences of *B*. *minutum* [38] showed that: *i*) the genome contains nearly twice as many gene families as that of its sister group, the apicomplexans; *ii*) only one-third of putative proteins encoded by the permanently condensed chromosomes of *B*. *minutum* have eukaryotic orthologs; *iii*) the identified genes are enriched in spliceosomal introns that use different recognition nucleotides at 5′ splice site; *iv*) genes are arranged in one direction throughout the genome, forming a cluster of genes, which was previously known only in trypanosomes, which are evolutionarily distantly related [82]. What’s more, a novel promoter element (motifs TTTT instead of the TATA box used by other eukaryotes) in the *F. kawagutii* genome, and a microRNA system potentially regulating gene expression in both symbiont and coral were observed [39].

The draft genomes provide an important resource for understanding the characteristics of *Symbiodiniaceae* biology and genetic differences between *Symbiodiniaceae* species (Figure 2). Global annotated genome analysis revealed that: *i*) common metabolic pathways in typical photosynthetic eukaryotes were found in the genomes of *F. kawagutii* and *C. goreaui* [7,39]; *ii*) *Symbiodiniaceae*-specific gene families, such as genes related to sexual reproduction, cyst formation, and germination and telomere synthesis, were identified in the assembled *F. kawagutii* and *B. minutum* genomes [39]; *iii*) some genes were highly expanded in the *Symbiodiniaceae* genome, including genes encoding chlorophyll a/b-binding proteins in *B. minutum* [38] and *Symbiodiniaceae* cladeA and *Symbiodiniaceae* cladeC [41], ion channel proteins and DNA repair/recombination proteins in all sequenced *Symbiodiniaceae* species [7,38,39,40,41,83], calcium channel and calmodulin families in *B. minutum* [38], heat shock proteins in *B. minutum* [38] and *F. kawagutii* [39], antioxidant genes, including the large thioredoxin gene family, superoxide dismutases and ascorbate peroxidases in *F. kawagutii* [39], as well as genes related to meiosis and response to light stress in *S. microadriaticum*, *C. goreaui,* and *F. kawagutii* [7]; *iv*) *Symbiodiniaceae* harbors a wide range of transport proteins [84], which are related to the supply of carbon and nitrogen [40], are responsive to reactive oxygen species and can prevent ultraviolet radiation [83], and all the studied dinoflagellates have more transmembrane transport proteins involved in the exchange of amino acids, lipids, and glycerol than other eukaryotes through the comparative analysis of genomes and transcriptomes [83]. Mycosporine-like amino acids (MAA) as specific ultraviolet radiation blockers protect the photosynthetic machinery of the dinoflagellate [28]. The MAA biosynthetic pathway involves dehydroquinate synthase, O-methyltransferase, an ATP (adenosine triphosphate)-grasp and non-ribosomal peptide synthetase [85]. It was found that the ability to synthesize mycosporine-like amino acids varied significantly among different *Symbiodiniaceae* clades [7,39,41]. The genome of *Symbiodiniaceae* (clade A) contains a gene cluster for the biosynthesis of mycosporine-like amino acids, which may be transferred from an endosymbiotic red alga [41]. In the genomes of *Symbiodiniaceae* (clade C) and *F. kawagutii*, early research suggested that they have completely lost the gene cluster [39,41]. However, a recent study using known proteins in bacteria, fungi, and cnidarians as queries, allowed the identification of putative genes encoding all five enzymes, including the short-chain dehydrogenase from the *S. microadriaticum*, *C. goreaui,* and *F. kawagutii* genomes [7].

The draft genomes also provide an important resource for understanding the molecular basis of coral-*Symbiodiniaceae* symbiosis. For example, *i*) genes related to synthesis and modification of amino acids were discovered in *F. kawagutii*, *C. goreaui*, *S. microadriaticum,* and *B. minutum* genomes, as many amino acids cannot be synthesized by coral host that must be supplied by *Symbiodiniaceae* [7,39]; *ii*) the recognition of *Symbiodiniaceae* by the hosts was considered to be mediated through the binding of *Symbiodiniaceae* high-mannose glycans by lectins on the coral cell surface [86], and the genes encoding a glycan biosynthesis pathway have been identified in *F. Kawagutii*, *C. goreaui*, *S. microadriaticum,* and *B. minutum* genomes [7,39]; *iii*) the cytosine methyltransferase gene family in *C. goreaui*, *S. microadriaticum,* and *B. minutum* genomes was shown to be expansive [7,39], which may partially explain a high proportion of diverse methylated nucleotides in dinoflagellates species [27]. Therefore, the observed decreases in fluorometrically assayed β-glucoronidase expression over time in *Amphidinium* and *Symbiodiniaceae* might suggest transgene silencing by a mechanism involving methylation [87]; *iv*) the new-found promoter elements and microRNA systems regulating gene expression may be useful as potential regulatory elements for metabolic engineering in dinoflagellates [39].

Besides whole genome sequencing, NGS can also be applied to other aspects of dinoflagellates biology research, such as mapping of structural rearrangements, analysis of DNA methylation, and identification of DNA-protein interactions [78]. Although NGS technologies are powerful, one of their major limitations is the relatively short reads they generated, which may lead to: *i*) mis-assemblies and gaps, as many repeated sequences that are longer than NGS reads exist in dinoflagellates genomes [88,89]; *ii*) more challenges to detect and characterize larger structural variations [90]. Another major limitation is that dinoflagellates genomes contain regions of extreme high content of guanine and cytosine, leading to inefficiently amplification by polymerase chain reaction [91]. The last is that a common NGS approach, bisulfite sequencing, can only detect C modification on treatment of DNA by bisulfite, but cannot distinguish 5mC and 5hmC [92]. In contrast, the emerging third-generation sequencing (TGS) technologies (e.g., single-molecule real-time sequencing used by PacBio, nanopore sequencing used by Oxford Nanopore Technologies and synthetic long reads used by Illumina/10X Genomics) can generate very long reads (tens of thousands of bases per nanopore) at the single-molecule level, which allows less sequencing bias and more homogeneous genome coverage [91]. In addition, Oxford Nanopore Technologies can directly detect DNA methylation 5mC and 6mA without amplification by polymerase chain reaction [93,94]. In spite of the major drawback, a high error rate (~15%), TGS technologies have been applied successfully in the analysis of repeated regions and structural variations, haplotype phasing, and transcriptome analysis [91]. In combination of NGS technologies, it is expected that TGS technologies have the potential to solve the sequencing bottleneck of dinoflagellates with large and complex genomes.

### 2.2. Transcriptomic Analysis of Marine Dinoflagellates

Although the enormous genome size of most dinoflagellates makes it challenging to obtain a whole genome sequence, one promising strategy to understand the function and regulation of both functionally known and unknown genes in an uncharacterized genome can be achieved by examining gene expression by transcriptomics analysis [95,96]. Several common methodologies for transcriptomics analysis used in dinoflagellates include: *i*) microarrays, a high throughput and relatively inexpensive method, which is based on hybridization but is subject to a number of limitations (e.g., relying upon existing genome sequence, high levels of background noise, and a limited dynamic range of detection of different isoforms and allelic expression) [97,98,99]; *ii*) expressed sequence tag (EST) sequencing, a method that is based on Sanger sequencing, and is relatively accurate but has many inherent disadvantages, such as low throughput, high cost, and the lack of the ability of quantification [97,100]; *iii*) massively parallel signature sequencing (MPSS), which is a tag-based high throughput method for sequencing millions of templates cloned on the surface of microbeads, is still subject to the limitations (e.g., high cost, a limited dynamic range of detection of isoforms, and disability of mapping a part of short tags to the reference genome) [97,101]; *iv*) RNA-Seq (RNA-sequencing), an high throughput approach employing NGS or TGS for direct sequencing of cDNA transcribed from the whole transcriptomes [97]. RNA-Seq shows clear advantages and has good application prospects, compared with other transcriptomic approaches [97]. First, it is not limited to the detection of transcripts that have corresponding genomic sequences, which attracts non-model organisms whose genome sequences have not yet been determined; second, it also has a high resolution to reveal the precise location of transcription boundaries, wide dynamic range to quantify gene expression level, and the ability to distinguish sequence variations, which are especially useful for complex transcriptomes [97]. Finally, RNA-Seq has a relatively low background, high accuracy of quantitative expression level, high technological reproducibility, and a much lower cost [97]. Thus, RNA-Seq technology has been most widely used approach in the transcriptomic analysis of dinoflagellates in recent years. A detailed summary on transcriptomics analysis of dinoflagellates is provided in Table 2.

Through transcriptomics analysis, several important aspects of genome structure and gene expression regulation in dinoflagellates have been discovered: *i*) transcriptomic analysis had recognized transcripts for all core histones (H2A, H2B, H3, H4) and their variants (H2A.X and H2A.Z) [125,130,131], despite the fact that dinoflagellates had been previously considered to completely lose histones genes [132]; *ii*) using the EST library, the genomic structures of 47 genes from the dinoflagellate *Amphidinium carterae* were identified and addressed [133]. The study showed that almost all highly expressed tags exist in large tandem gene arrays with short intergenic spacers, while the second class of genes showed high intron density and significantly lower copy number. A polyadenylation signal was also discovered in genomic copies at the exact polyadenylation site and was conserved between species; *iii*) the presence of 5′- *trans*-spliced leader addition in mRNA processing was found in several dinoflagellates using EST data, suggesting that dinoflagellates heavily rely on the posttranscriptional regulation of gene expression [131,133,134,135]; *iv*) compared to other eukaryotes, a reduced role for the transcriptional regulation of gene expression was discovered. For example, the analysis showed that only about 3% of the genes were significantly changed at transcriptional level in response to a circadian clock [126] and only 4% in response to oxidative stress in *Pyrocystis lunula* [125].

Transcriptome profiling also allows for the study of the metabolic and physiological response of dinoflagellates responsive to various stresses. As summarized in Table 2, transcriptional changes of dinoflagellates under different growth conditions have been extensively studied. First, for some species of dinoflagellates responsible for forming HABs, studies have been conducted on how environmental conditions conductive to HABs [46,105,109,114,115] affect the expression of toxin related genes and the toxin biosynthesis pathway [47,56,60,108] as well as physiological processes of HABs outbreak [107]. For example, in *Alexandrium tamarense*, MPSS results showed that transcripts of chlorophyll a/b binding protein, histone family protein, S-adenosylmethionine synthetase, and S-adenosylhomocysteine hydrolase were the highest expressed among all tested conditions that provoke the formation of HABs [46]. Microarray-based comparative transcriptome profiles of dinoflagellate *Alexandrium minutum* suggested that many genes were more highly expressed in the toxic than in the non-toxic strain, and several genes were even expressed only in the toxic strains [47]. In addition, a variety of metabolic pathways specially related to various N (cyanate, urea, nitrate/nitrite, and ammonium) uptake and assimilation were enriched, which is likely to confer competitive advantages for bloom formation or maintenance in *Alexandrium catenella* [107]. Using time sequential metatranscriptomic, they analyzed a natural assemblage that evolved from diatom (*Skeletonema*) dominance to a dinoflagellate (*Prorocentrum donghaiense*) bloom [136]. The results showed that during the dominant period, a similar series of most active metabolic processes (energy and nutrient acquisition, stress resistance) promoted growth and distinct metabolic pathways were used by diatom and dinoflagellates in their respective dominance, while *P. donghaiense* possessed more diversified light energy and phosphate acquisition strategy and antimicrobial defense, which might cause them to grow faster than diatoms and form blooms. Second, due to the potential significance in coral bleaching, transcriptional changes of *Symbiodiniaceae* in response to thermal stress or light have been widely investigated [64,66,67,121,122]. Transcriptomic responses to heat stress in *F. kawagutii* were determined, and the results showed that 357 genes were differentially expressed under heat stress, and most of them were involved in regulating cell wall modulation and the transport of iron, oxygen, and major nutrients. What’s more, the expression of heat shock proteins was strongly elevated during heat stress, within expectations. The results also showed that the demand for nutrients, iron, and oxygen in *F*. *kawagutii* might be higher under heat stress [122]. When grown under autotrophic or mixotrophic conditions, the expression levels of many *Symbiodiniaceae* SSB01 genes seemed to be significantly affected by light, such as a cryptochrome 2 gene (declined in high light), regulators of Chromatin Condensation (RCC1) (declined in the dark), a light harvesting AcpPC protein (increased in high light autotrophic conditions), and several cell adhesion proteins (rapidly declined when the culture was moved from low light to darkness) [66]. The increased transcript level of AcpPC gene suggests that it is involved in photo-protection and the dissipation of excess absorbed light energy. The decreased cell adhesion protein level is related to the significant change in cell surface morphology, which may reflect the complexity of the extracellular matrix [66]. These results provided valuable insights into the molecular basis of thermal or light resistance of dinoflagellates and coral bleaching. Third, to characterize the mechanisms during DHA production, a transcriptional analysis of the heterotrophic *C. cohnii* under different cultivation stages and conditions was carried out. By analyzing and comparing the differential gene expression profiles during lipid accumulation and DHA formation stages, core metabolism pathways in *C. cohnii* were proposed (Figure 3) [72]. In addition, transcripts related to fatty acid biosynthesis, starch and sucrose metabolism, and unsaturated fatty acids biosynthesis were significantly up-regulated during late-stage fermentation [72]. The study also found that some polyketide synthases and fatty acid desaturases may be involved in the biosynthesis of DHA. In addition, some enzymes involved in reducing power metabolism, such as malic enzyme and isocitrate dehydrogenase were up-regulated 1.7- to 2.3-fold during the lipid and DHA accumulation stages respectively, while the transcript of glucose-6-phosphate 1-dehydrogenase was downregulated 0.86- to 0.57-fold, suggesting that *C. cohnii* might use the malic enzyme and isocitrate dehydrogenase instead of glucose-6-phosphate 1-dehydrogenase to produce NADPH (nicotinamide adenine dinucleotide phosphate) [137]. Furthermore, *C. cohnii* mutants with high DHA productivity were also examined through transcriptomics analysis [12]. It was found that gene expression levels involved in fatty acid biosynthesis, energy, central carbohydrate, and amino acid metabolism were upregulated in the mutant with high DHA productivity, compared to the wild type. These results provide a basis for understanding for improving lipid accumulation and DHA production by rational engineering of *C. cohnii* in the future.

Although transcriptomic analysis provides a ‘‘snap shot’’ of gene expression under specific environmental and physiological conditions, it was also found that mRNA expression levels sometimes have a poor correlational relationship with the phenotype in dinoflagellates. For example, oscillating RNAs were not detected over the circadian cycle of the dinoflagellate *Lingulodinium polyedrum* [138]. They also discovered that the timing of the bioluminescence and photosynthesis rhythms remained unchanged, even when transcription rates had decreased to about 5% of the levels of untreated cells with the addition of inhibitors actinomycin D and cordycepin. Furthermore, there were no detectable changes in gene expression across the two types of *Symbiodiniaceae* (one was thermotolerant type and the other was more susceptible type) under thermal stress, even as the symbiosis was breaking down [64]. Some of these phenomena could be partially due to many cellular mechanisms acting at a post-transcriptional level. Another important feature is the generally low degree of congruency between mRNA and protein expression. For example, for the 167 proteins in *Symbiodiniaceae* downregulated at variable temperatures, only two corresponding mRNAs were differentially expressed between different treatments, while for 378 differentially expressed genes, none of their corresponding proteins was differentially expressed [68]. Therefore, it is worth noting the inherent risk of inferring cellular behavior based on transcriptional data alone. More and integrated omics or physiological analysis should be carried out for the full confirmation of metabolic and physiological behaviors of dinoflagellates.

### 2.3. Proteomic Analysis of Marine Dinoflagellates

As the main component of cellular structure and communication, protein is believed to be a more relevant indicator of an observed cellular phenotype than RNA [139]. Two major strategies for the separation and visualization of proteins are applied in proteomics: *i*) the two dimensional gel electrophoresis (2-DE) strategy, which separates proteins based on their masses and isoelectric points, followed by mass spectrometric identification; *ii*) gel-free profiling procedures, which rely on multidimensional separations coupling micro-scale separations with automated tandem mass spectrometry [140,141]. In the past decades, the proteomics technology has been rapidly developed, which make it possible to more directly probe the cellular behaviors of dinoflagellates than before [26,142].

Gel-free proteomics techniques, such as matrix-assisted laser desorption ionization-time-of-flight mass spectrometry (MALDI-TOF-MS) [143] and capillary liquid chromatography followed by tandem mass spectrometry (LC-MS) [144], have been applied to dinoflagellates proteomes. In particular, a non-gel based quantitative proteomic method named iTRAQ (isobaric Tags for Relative and Absolute Quantification), which quantifies proteins based on peptide labeling and allows large-scale identification and accurate quantification of proteins from multiple samples within broad dynamic ranges of protein abundance, have been conducted for *A. catenella* [58] or *Karlodinium veneficum* [145]. In the study, a total of 3488 or 4922 proteins were successfully identified from the proteomics of *A. catenella* or *K. veneficum* respectively, which are the highest number of proteins identified in dinoflagellates. Comparing the protein profiles of a toxin-producing dinoflagellate *A. catenella* (ACHK-T) and its non-toxic mutant (ACHK-NT) using a combination of iTRAQ-based proteomic approach and a transcriptomic database, they found different carbon and energy utilization strategies between ACHK-T and ACHK-NT, and discovered seven cyanobacterial toxin-related proteins and *sxtG* of dinoflagellates, which were identified as candidates involved in toxin biosynthesis but had no obvious difference between the two strains [58]. In *K. veneficum*, the changes in non-photochemical quenching and molecular mechanism under phosphorus deprivation were studied [145]. Proteomics results based on iTRAQ showed that non-photochemical quenching in *K. veneficum* increased significantly under phosphorus deprivation. Correspondingly, three light-harvesting complex stress-related proteins and energy production- and conversion-related proteins were up-regulated, while many proteins related to genetic information flow were down-regulated. It is expected that the iTRAQ-based proteomic approach could be utilized more frequently in future work as increasing genomic information on the dinoflagellates appears. However, proteins identified by iTRAQ-based proteomic approach are still far fewer than genes identified by genomics and transcriptomics approaches. One of the important reasons for this is the tedious and immature protein extraction procedures (including cell wall disruption and protein extraction), as most dinoflagellates have a tough and complex cell cortex leading to incomplete cell wall breakage and chemical contamination during extraction, which can interfere protein analysis [142]. Another important reason is protein identification, which is detailed below.

Proteomics has shown its powerful utilization in exploring the physiological and metabolic characteristics of dinoflagellates. Proteomics analysis has been used to identify new dinoflagellates species [48,143,146,147,148,149]. Traditional morphological approaches have been widely applied in dinoflagellate species identification, but these methods exhibit weaknesses in distinguishing closely related species with similar morphologies [26]. Using two-dimensional gel electrophoresis (2-DE) proteomics technology, species-specific protein expression profiles and standards of species identification in ten dinoflagellate species were established [147]. 2-DE has also been used to differentiate morphospecies (toxic and nontoxic) of *A*. *minutum*, and differentially expressed proteins between different morphospecies were observed at the proteome level [143]. Proteomics has also been employed to discover proteins of *Symbiodiniaceae* [150] and to reveal proteins involved in symbiosis and responses to environmental stress [151]. Using proteomics based on LC–MS/MS, 417 protein spots were identified in the endosymbiotic zooxanthellae from *Euphyllia glabrescens* and three marker proteins (green fluorescent protein R7, Histone H2B, and peridinin chlorophyll-a binding protein) were also found [150]. A total of 8098 MS/MS spectra relevant to peptides from the fraction of endosymbiont were identified, while only 26 peptides showed a significant change when treated with thermal stress or ion limitation. Surprisingly, the expression levels of proteins related to antioxidant or heat stress phenotypes were roughly the same, while proteins involved in protein biosynthesis were highly expressed. Proteomics has also been used to compare the performance of different *Symbiodiniaceae* species [144,152]. Using LC-MS based proteomics, they compared proteomes of the model sea anemone *Exaiptasia pallida* colonized by different dinoflagellate symbiont (*B. minutum* and *Durusdinium trenchii*) [144]. Results showed that in anemones containing *D. trenchii* (heterologous symbiont), Niemann-Pick C2 proteins, and glutamine synthetases were lowly expressed, while methionine-synthesizing betaine–homocysteine S-methyltransferases and proteins with predicted oxidative stress response functions were highly expressed when compared with anemones containing *B. minutum* (homologous symbiont). In another study, the researchers found that high-density symbiotic colonization required avoiding the immune responses of host cells, enhancing ammonium regulation, and inhibiting the phagocytosis of host cells after colonization [152]. *iv)* Proteins related to toxin biosynthesis were identified in *A*. *catenella* through proteomics analysis [153]. The proteomic results showed that the abundance of nine proteins with known functions in paralytic shellfish toxin-producing cyanobacteria was differentially regulated between toxin biosynthesis stages, suggesting that they might be involved in toxin biosynthesis in *A. catenella*.

In spite of the progress made, proteomics analysis of dinoflagellates still faces some technical challenges, especially for protein identification and validation. Protein identification largely depends on relevant genomic sequences, but so far only six draft whole genome sequences of dinoflagellates are available, and transcriptomes are still insufficient as a substitute for the query due to low coverage. In addition, peptides often contain post-translational modifications that lead to mass change of the fragments, thus hindering identification. Nevertheless, with the rapid development of genomics and transcriptomics in the near future, it is expected that proteomics will become a more powerful tool for studying dinoflagellates biology.

### 2.4. Metabolomic Analysis of Marine Dinoflagellates

Metabolomics is a global interrogation of cellular components, and has been gradually considered as a crucial supplement to other omics in exploring the mechanisms of dinoflagellates biology [154]. The advantages of metabolomics lie in its ability to detect qualitative and quantitative changes in a large number of metabolites using mass spectrometry and nuclear magnetic resonance spectroscopy [154]. Although there are a number of high-resolution technologies capable of detecting and identifying changes in metabolite profiles, our understanding of how these differently expressed metabolites affect overall biological function is still in infancy [155,156,157]. Currently, gas chromatography time-of-flight mass spectrometry (GC-TOF-MS), high-performance liquid chromatography mass spectrometry (LC-MS), and ^13^C-labeling based metabolic flux analyses have been mainly used in dinoflagellates metabolomics analysis (Table 3) [69,73,158,159].

Although the application of metabolomics to dinoflagellates is relatively new, it has already made significant contributions to dinoflagellates biology research (Table 3): *i*) metabolomics studies focusing on lipogenetic mechanisms of *C. cohnii* have been conducted. For example, metabolomic analysis has been used to identify metabolic modules and hub metabolites related to the positive role of the antioxidant butylated hydroxyanisole on lipogenesis in *C. cohnii* [13], to determine mechanisms relevant to glucose tolerance through adaptive laboratory evolution [11], the responsive metabolites associated with varying dissolved oxygen levels [74], as well as the metabolic changes among different nitrogen feeding fermentation conditions in *C. cohnii* [76]; *ii*) metabolite profiles related to several growth conditions of marine symbiotic dinoflagellates of *Symbiodiniaceae* were also examined using GC-MS based metabolomic analysis [161], and the results showed that both the production of sterols and sugars and the abundance of hexose and inositol were different between different *Symbiodiniaceae* species. As acidification markedly inhibited *B. minutum* growth [164], the responsive metabolite profile of *B. minutum* to acidification was studied [162]. The results showed that: *a*) saturated fatty acids and oligosaccharides were accumulated when *B. minutum* was cultured in the acidification condition, which was considered as an important strategy to adapt to acidification; and *b*) the inhibition of the growth rate was possibly due to the affection of acidification to the biosynthesis of amino acids and proteins of *B. minutum*; *iii*) ^13^C labelling coupled with GC-MS was used to track carbon changes in metabolic flux dinoflagellates. For example, autotrophic carbon fate was mapped in both Aiptasia and *Symbiodiniaceae* (a model cnidarian-dinoflagellate symbiosis) exposed to thermal stress [69]. A newly fixed carbon fate in the model cnidarian *Exaiptasia pallida* was characterized with colonized with either native *B. minutum* or non-native *D. trenchii* using ^13^C-labeling coupled with GC-MS [159]. Different abundance and diversity of metabolites were detected between anemones colonized with different *Symbiodiniaceae* species and significant alterations to host molecular signaling pathways were also revealed. For heterotrophic *C. cohnii*, ^13^C-labeling based metabolic flux analysis has been used to explore the central flux distribution after addition of ethanolamine to stimulate lipid accumulation [73]. It was found that activity of the glycolysis pathway and citrate pyruvate cycle was increased, while that of the pentose phosphate pathway and TCA cycle was attenuated after adding ethanolamine; *iv*) exometabolomics, which focused on the research of complete small molecules cells secret into their environments [165], was also used for investigating interactions of different dinoflagellates species. For example, when *Ostreopsis* cf. *ovata* was co-cultivated with diatom *Licmophora paradoxa*, the exometabolome of *L. paradoxa* was found impair the growth and the photochemistry of *Ostreopsis* cf. *ovata* in both bioassays and co-cultures, and some biomarkers possibly involved in the inhibition effects were later identified using a metabolomic approach [61].

Although the comprehensiveness and measurement accuracy of metabolomics still needs further improvements [155,156,157], early research has demonstrated that metabolomics is a powerful tool in exploring metabolism and filling the phenotype-genotype gap because it presents a closer picture of cellular activity than other omics methods [157]. Furthermore, metabolomics is considered as a more promising approach for interpreting the basic characteristics of dinoflagellates without sequencing entire genomes.

### 2.5. Integrated Omics Analysis of Marine Dinoflagellates

In recent years, it has been realized that single “omics” analysis is not sufficient for characterizing the complexity of dinoflagellate biological systems. Therefore, integration of multiple “omics” approach is required to obtain a relatively precise picture of dinoflagellates. Some attempts have been made recently to integrate heterogeneous “omics” datasets in various microbial systems, and the results have demonstrated that the “multi-omics” method is a powerful tool for understanding the functional principles and dynamics of total cellular systems [37,166,167,168,169,170].

The integration of the “multi-omics” approach was also applied to explore complex metabolic networks and global regulatory mechanisms in dinoflagellates [68,71,72,171]. For example, *i*) integrated transcriptomic and metabolomic analysis was applied to characterize the molecular and physiological processes of DHA synthesis in the fed-batch fermentation of *C. cohnii*, and differently expressed genes involved in fatty acid and DHA biosynthesis were discovered [72]; *ii*) integrated transcriptomic and metabolomic analysis was also used to compare the effects on the sea anemone *Exaiptasia pallida* when colonized by a homologous symbiont *B. minutum* and a heterologous, opportunistic, and thermally tolerant symbiont *D. trenchii* [171]. It was shown that the catabolism of fixed carbon storage, metabolic signals, and immune processes were increased in *E*. *pallide* accompanied with heterogenous symbiosis *D. trenchii*, in comparison with homologous *B. minutum* colonized hosts; *iii*) integrated transcriptomic and proteomic analysis was used to study the temperature adaptation mechanism of Zooxanthellate cnidarians (the Red Sea (RS), North Carolina (CC7), and Hawaii (H2)) [71]. A common core response to thermal stress (24 h at 32 °C), containing protein folding and oxidative stress pathways, was highlighted through comparisons. The level of antioxidant gene expression was increased in all three anemones, while RS anemones showed the greatest increase. Reactive oxygen species production was different between three strains, which were symbiont-driven, while RS anemones showed significantly lower levels *in hospite* [68].

When using the multi-omics approach, there still are many tough challenges. For example, low correlation between the transcript and protein levels, which might be due to posttranscriptional regulation of gene expression or a lack of the proper statistical tools for biological interpretation [37]. In addition to the risk of using single omics analysis, these is also a need to develop new bioinformatics tools and improve the availability of public data repositories.

## 3. Conclusions and Prospective Research

Omics research on dinoflagellates has attracted great interest in the past few decades, as it has substantially contributed to the better understanding of the dinoflagellates at the molecular level. However, there are still challenges in omics analysis, and the risks of only using one kind of omics technology to explore dinoflagellates biology deserve attention.

In future studies, several aspects of omics research on dinoflagellates should be strengthened. First, more attention should be paid to the complete genome sequencing of dinoflagellates. The newly developed DNA sequencing technologies, such as single-molecule DNA sequencing, could compensate for the shortcomings of NGS for this purpose, by acting as long scaffolds to resolve highly repetitive genomic regions [172], and make whole-genome sequencing of dinoflagellates much more feasible. Second, strengthening the description of the proteome/metabolome of a dinoflagellate, and establishing a reference proteome/metabolome of a dinoflagellate. These would will greatly increase the rate of discovery in dinoflagellate biology and reveal biomarkers for certain functional characteristics or responses of dinoflagellates. Third, integrated multi-omics analysis is important for exploring dynamic process spanning multiple cellular components in dinoflagellates [37]. Functional genomics data sets, such as transcriptomics, proteomics, and genome-wide mutant screens, can provide additional layers of gene-specific functional data. This integrated approach can minimize errors during analysis and the result will be more comprehensive and accurate. Fourth, it is urgent that the mechanism of HABs formation and toxin biosynthesis receive intensive study. In recent years, there has been a negative influence of dinoflagellate blooms on the environment and human society, as they are occurring more frequently and becoming larger in scale [173]. Finally, as transformation systems for selective species of dinoflagellates, such as *Amphidinium*, *S. microadriaticum*, *Symbiodiniaceae,* and *C. cohnii*, have been reported [75,87,174], the results obtained from omics analysis can provide more targets and pathways to accelerate the process of genetic engineering in dinoflagellates. More mutants with superior characteristics, such as *C. cohnii,* with higher fractions of DHA, or *Symbiodiniaceae,* with the ability to tolerate higher temperatures will be constructed.

## Figures and Tables

**Figure 1 microorganisms-07-00288-f001:**
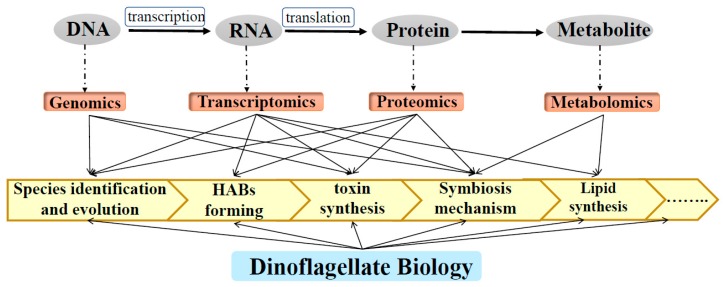
Application of omics technologies for dinoflagellate biology research.

**Figure 2 microorganisms-07-00288-f002:**
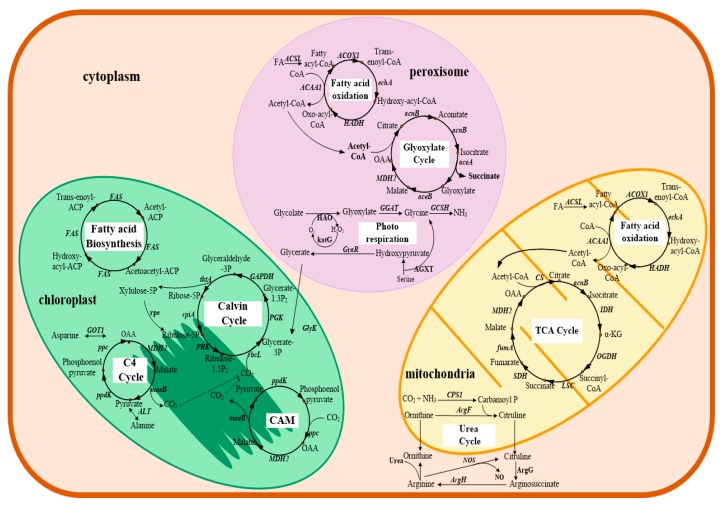
Major metabolic pathways in *F. kawagutii*. Complete pathways for the mitochondrial tricarboxylic acid cycle (TCAcycle), fatty acid oxidation and the urea cycle, the chloroplastic Calvin cycle, dicarboxylic acid cycle (C4 cycle), crassulacean acid metabolism (CAM), fatty acid biosynthesis, peroxisomal fatty acid oxidation, glyoxylate cycle, and photorespiration are found in the *F. kawagutii* genome. *FAS*, fatty acid synthase; *GOT1*, aspartate aminotransferase; *ppc*, phosphoenolpyruvate carboxylase; *MDH2*, malate dehydrogenase; *maeB*, malic enzyme; *ALT*, alanine transaminase; *ppdK*, pyruvate, phosphate dikinase; *rpe*, L-ribulose phosphate epimerase; *tktA*, transketolase; *GAPDH*, glyceraldehyde 3-phosphate dehydrogenase; *PGK*, phosphoglycerate kinase; *rbcL*, ribulose-bisphosphate carboxylase large chain; *PRK*, phosphoribulokinase; *rpiA*, ribose 5-phosphate isomerase A; *glyK*, glycerate kinase; *ACSL*, long-chain acyl-CoA synthetase; *ACOX1*, acyl-CoA oxidase; *echA*, enoyl-CoA hydratase; *HADH*, 3-hydroxyacyl-CoA dehydrogenase; *ACAA1*, acetyl-CoA acyltransferase; *acnB*, aconitate hydratase; *aceA*, isocitrate lyase; *AceB*, malate synthase; *HAO*, (S)-2-hydroxy-acid oxidase; *katG*, catalase; *GGAT*, glutamate--glyoxylate aminotransferase; *GCSH*, glycinSe cleavage system H protein; *GyaR*, glyoxylate reductase; *AGXT*, alanine-glyoxylate transaminase / serine-glyoxylate transaminase / serine-pyruvate transaminase; *CS*, citrate synthase; *IDH*, isocitrate dehydrogenase; *OGDH*, α-ketoglutarate dehydrogenase; *LSC*, Succinyl-CoA synthesase; *SDH*, succinate dehydrogetase; *fumA*, fumarase; *CPS1*, carbamoyl phosphate synthetase I; *argF*, ornithine carbamoyltransferase; *argG*, argininosuccinate synthase; *argH*, argininosuccinate lyase; *NOS*, NO synthetase; FA, fatty acid; OAA, Oxaloacetic acid; α-KG, α-ketoglutarate. (Schemed based on the study [38]).

**Figure 3 microorganisms-07-00288-f003:**
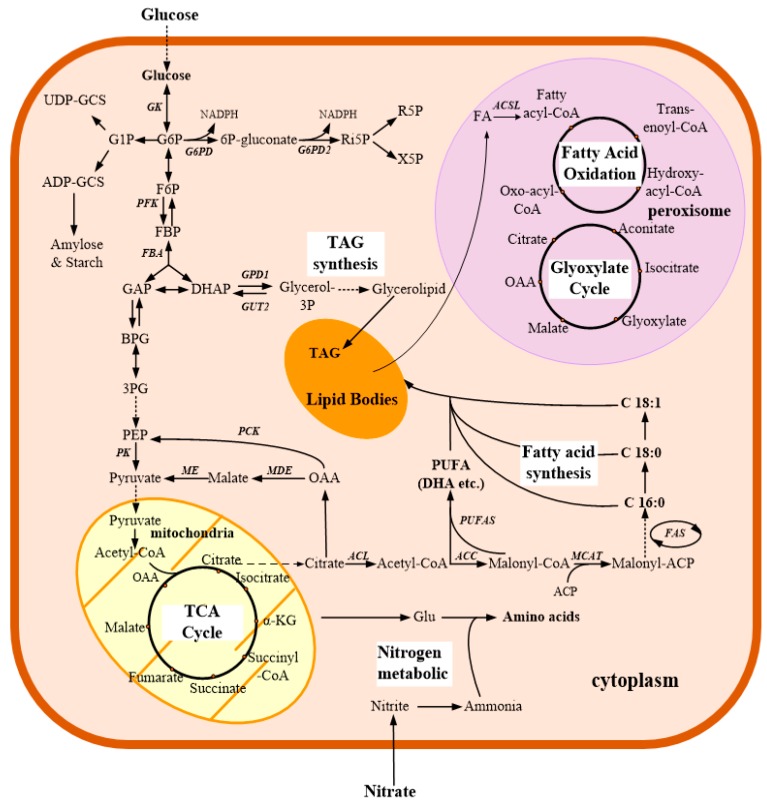
Metabolic pathways in *C*. *cohnii*. Pathways associated with central carbohydrate, fatty acid and TAG biosynthesis, fatty acid oxidation, glyoxylate cycle, and nitrogen metabolic pathways. *GK*, glucokinase; G6PD, glucose-6-phosphate 1-dehydrogenase; G6PD2, 6-phosphogluconate dehydrogenase; PFK, 6-phosphofructokinase; FBA, fructosebisphosphate aldolase; GPD1, glycerol-3-phosphate dehydrogenase; GUT2, glycerol-kinase; PCK, phosphoenolpyruvate carboxykinase; PK, pyruvate kinase; ME, malic enzyme; MDE, malate dehydrogenase; ACL, ATP citrate lyase; ACC, acetyl-CoA carboxylase; *PUFAS*, polyunsaturated fatty acid synthase; *FAS*, fatty acid synthase; *MCAT*, malonyl-CoA: ACP transacylase; *ACSL*, long-chain acyl-CoA synthetase; DHA, docosahexaenoic acid; PUFA, polyunsaturated fatty acid; TAG, triacylglycerol. (Schemed based on the study [72]).

**Table 1 microorganisms-07-00288-t001:** Genomic summary of six dinoflagellates draft genomes.

		*Breviolum minutum* [38]	*Fugacium kawagutii* CCMP2468 [39]	*Symbiodinium microadriaticum* CCMP2467 [40]	*Cladocopium goreaui* SCF055-01 [7]	*Fugacium kawagutii* CCMP2468 [7]	*Symbiodiniaceae* [41]	*Symbiodiniaceae* [41]
Clade		B	F	A	C	F	A	C
	Total assembled length (bp)	615,520,517	935,067,369	808,242,489	1,027,792,016	1,048,482,934	766,659,703	704,779,698
	G+C content (%)	43.6	43.97	50.51	44.83	45.72	49.9	43.0
Genes	Number of genes	41,925	36,850	49,109	35,913	26,609	69,018	65,832
	Mean length of genes (bp)	11,959	3788	12,898	6967	6507	8834	8192
	Mean length of transcripts (bp)	2067	1041	2377	1766	1736	1423	1479
Exons	No. of exons per gene	19.6	4.1	21.8	10	8.7	13.38	11.27
	Mean length (bp)	99.8	256	109.5	175.9	199.5	105	130
	Total length (Mb)	82.1	38.4	117.3	63.4	46.2	98.2	97.3
Introns	No. of genes with introns (%)	95.3	64.1	98.2	92.9	94	83.4	80.3
	Mean length (bp)	499	893	504.7	575.1	619.4	561	622
	Total length (Mb)	331.5	101.2	516.1	186.8	126.9	481.8	421.2
Intergenic regions	Mean length (bp)	2064	17,888	3633	10,627	23,042	2008	2202
platform		Roche 454 GS-FLX and Illumina (GAIIx)	Illumina HiSeq 2000	Illumina HiSeq	Illumina HiSeq 2500	Illumina HiSeq 2500	Illumina (GAIIx) and Hiseq	Illumina (GAIIx) and Hiseq
Bioproject ID		PRJDB732	SRA148697	PRJNA292355	PRJEB20399	PRJEB20399	PRJDB3242	PRJDB3243

GAIIx: Genome Analyzer IIx.

**Table 2 microorganisms-07-00288-t002:** Transcriptomics studies of marine dinoflagellates.

Species	Propose	Platform	Ref./year
Dinoflagellates responsible for HABs and toxic			
*Karenia brevis*	To establish a database of *K. brevis* ESTs and identify conserved eukaryotic genes	microarray	[102] 2005
*Karenia brevis*	To compare gene expression in response to light and dark	microarray	[103] 2007
*Karenia brevis*	To investigate characterization and expression of nuclear-encoded polyketide synthases (PKSs)	microarray	[104] 2010
*Karenia brevis*	To address transcriptional responses to nitrogen and phosphorus depletion and addition	microarray	[105] 2011
*Karenia brevis*	To compare global transcriptome changes that accompany the entry and maintenance of stationary phase up to the onset of cell death.	microarray	[106] 2012
*Alexandrium catenella*	To study the content and regulation of the dinoflagellate genome	MPSS	[44] 2006
*Alexandrium catenella* CCMP1719	To generate time-serial ESTs throughout a diel cycle during bloom	Roche 454 GS FLX	[107] 2015
*Alexandrium minutum*	To compare gene expression in toxic versus non-toxic strains	Microarray	[47] 2010
*Alexandrium minutum*	To determine transcriptional changes during the copepod-provoked induction of higher toxicity in *A. minutum.*	microarray	[108] 2011
*Alexandrium tamarense* CCMP1598	To investigate global transcriptional regulation under four different conditions, with xenic, nitrogen-limited, phosphorus- limited, and nutrient-replete	MPSS	[46] 2010
*Alexandrium monilatum* CCMP3105	To study transcriptional responses to limiting N and P conditions	Illumina HiSeq 2000	[109] 2017
*Alexandrium ostenfeldii*	To analyze gene composition, and structure and peculiarities of gene regulation		[110] 2011
*Gambierdiscus polynesiensis*	To reveal the mechanisms of CTX biosynthesis using transcriptomics	Roche 454 GS FLX	[54] 2014
*Alexandrium catenella*	To construct an expressed sequence tag (EST) library from *Alexandrium catenella*	*	[43] 2008
*Alexandrium catenella*	To determine the gene repertoire based on (NGS) technologies	Illumina Genome Analyzer.	[55] 2014
*Alexandrium catenella*	To study the mechanism of PSTs synthesis using transcriptome profiles of a toxin-producing and its non-toxic mutant form	Illumina Hiseq 2000	[56] 2014
*Alexandrium catenella* (ACHK-T)	To study molecular mechanisms for PST biosynthesis using the transcriptome profiles of a toxin-producing dinoflagellates at different toxin biosynthesis stages within the cell cycle	Illumina Hiseq 2000	[60] 2017
*Amphidinium carterae*	To study de novo transcriptome for the identification of enzymes with biotechnological potential	Illumina HiSeq.1000	[111] 2017
*Azadinium spinosum*	Transcriptomic and genomic characterization of the toxigenic dinoflagellate with emphasis on polyketide synthase genes	Roche 454 GS FLX	[112] 2015
*Prorocentrum minimum* CCMP 1329	To identify genetic modules mediating the Jekyll and Hyde Interaction	Illumina HiSeq 2500	[113] 2015
*Prorocentrum minimum* D-127	To evaluate genome-scale responses when exposed to polychlorinated biphenyl	microarray	[114] 2018
*Cochlodinium polykrikoides*	To compare transcriptional responses to the algicide copper sulfate	Illumina HiSeq 2500	[115] 2016
*Karenia mikimotoi* C32-HK	To reveal non-alkaline phosphatase-based molecular machinery of ATP utilization	Roche 454 GS FLX	[116] 2017
Dinoflagellates responsible for HABs and nontoxic			
*Scrippsiella trochoidea* CCMP 3099	To study the biochemical and physiological adaptations related to nutrient depletion	Illumina HiSeq 2000	[117] 2016
Dinoflagellate producing ciguatoxin			
*Gambierdiscus caribaeus*	To trace the evolutionary history of C and N pathways in this phylum using transcriptome data	Illumina MiSeq	[118] 2016
Dinoflagellates responsible for diarrheic shellfish poisoning			
*Prorocentrum lima* CCMP 2579	To compare the molecular and cellular responses to N limitation	Illumina HiSeq 2500	[119] 2018
Symbiotic dinoflagellates			
*Symbiodiniaceae* (clade A) *Symbiodiniaceae* (clade B)	To construct an EST dataset for the genetic study of *Symbiodiniaceae*	Roche 454 GS FLX	[63] 2012
*Symbiodiniaceae* Type D2 *Symbiodiniaceae* Type C3K	To compare transcriptional responses to thermal stress and the differences among physiologically susceptible and tolerant types	Illumina HiSeq	[64] 2014
*Symbiodinium microadriaticum* clade A1, CCMP 2467	To study the repertoire of endogenous smRNAs and to identify potential gene targets in dinoflagellates	Illumina HiSeq 2000	[120] 2013
*Symbiodiniaceae* SSB01	To study transcriptional responses to immediate changes in light intensity when grown under autotrophic or mixotrophic conditions	Illumina HiSeq 2000	[66] 2015
*Symbiodiniaceae* type C1	To study physiological and transcriptional responses to heat stress and to identify the gene related thermal response	Illumina HiSeq 2500	[67] 2016
*Symbiodiniaceae* type C1	To study effects of viral infections to *Symbiodiniaceae* when heat-stressed	Illumina HiSeq 2500	[70] 2017
*Symbiodiniaceae* (clade F)	To study the transcriptional response of cellular mechanisms under future temperature conditions	Illumina HiSeq 2000	[121] 2017
*Fugacium kawagutii* CCMP2468	To study transcriptional responses to thermal stress and varied phosphorus conditions	*	[122] 2019
*The fastest swimming dinoflagellates*			
*Ansanella granifera*	To study the structural and functional genes of dinoflagellate flagelle	Illumina HiSeq 2500	[123] 2017
high DHA yields dinoflagellate			
*Crypthecodinium cohnii* ATCC 30556	To compare transcriptional differences on a high lipid producing mutant with the wide-type strain	Illumina HiSeq 4000	[12] 2017
*Crypthecodinium cohnii* ATCC 30556	To compare transcriptional difference following the growth course during fed-batch fermentation	Illumina HiSeq 2500	[72] 2017
*Basal dinoflagellates*			
*Perkinsus olseni*	To study distribution and evolution of peroxisomes in the super ensemble Alveolata	MiSeq and HiSeq 2000	[124] 2017
*Pyrocystis lunula*	To study gene expression in *Pyrocystis lunula* in responsive to the addition of sodium nitrite and paraquat	microarray	[125] 2003
*Pyrocystis lunula*	To analyze circadian regulation at transcriptional levels in *Pyrocystis lunula*	microarray	[126] 2003
*Lingulodinium polyedrum*	To study molecular underpinnings of cold-induced cyst formation Sin the dinoflagellate *L. polyedrum*	Illumina HiSeq	[127] 2014
*Oxyrrhis marina*	To study transcriptional responses to salinity		[128] 2011
*Dinophysis acuminata*	To determine whether the dinoflagellates contain nuclear-encoded genes for plastid function	Roche 454 GS FLX	[129] 2010

*, not mentioned; HABs: harmful algal blooms; ESTs: expressed sequence tags; MPSS: Massively Parallel Signature Sequencing; CTX: Ciguatoxins; NGS: next-generation sequencing; EDCs: Endocrine disrupting chemicals; PST: paralytic shellfish toxins.

**Table 3 microorganisms-07-00288-t003:** Metabolomics studies of marine dinoflagellates.

Species	Propose	Methods	Ref./year
*Symbiodiniaceae* (clade B)	To analysis fatty acid composition in cellular TAGs	GC-MS	[160] 2014
*Symbiodinium microadriaticum*, *Breviolum minutum*, *Breviolum psygmophilum*, *Durusdinium trenchii*,	To study the metabolite profile of marine symbiotic dinoflagellates of *Symbiodiniaceae*	GC-MS	[161] 2015
*Symbiodiniaceae*	To compare widespread change in carbon fate during coral bleaching	^13^C labelling coupled to GC-MS	[158] 2017
*Symbiodiniaceae*	To map carbon fate during bleaching	^13^C labelling coupled to GC-MS	[69] 2017
*Breviolum minutum*	to determine metabolomic changes of *Breviolum minutum* in acidification condition, and explore the possible mechanisms	LC-MS/MS	[162] 2019
*Crypthecodinium cohnii* ATCC 30772	To study mechanism of antioxidant butylated hydroxyanisole on lipid accumulation	LC-MS and GC-MS	[13] 2014
*Crypthecodinium cohnii* ATCC 30556	To reveal mechanisms related to glucose tolerance of *C. cohnii* through adaptive laboratory evolution	GC-MS	[11] 2017
*Crypthecodinium cohnii* ATCC 30556	To study physiological metabolism of *C. cohnii* for increased DHA production	GC-MS	[72] 2017
*Crypthecodinium cohnii* ATCC 30556	To gain understanding of the lipid metabolism and mechanism for the positive effects of modulator ethanolamine	^13^C labelling coupled to GC-MS	[73] 2018
*Crypthecodinium cohnii* ATCC 30556	To study metabolic responses to different dissolved oxygen levels during DHA fermentation	GC-MS	[74] 2018
*Crypthecodinium cohnii* ATCC 30556	To compare molecular mechanisms of lipid accumulation in different strains	LC-MS and GC-MS	[75] 2018
*Crypthecodinium cohnii* M-1-2	To determine the metabolic changes under different nitrogen feeding conditions	GC-MS	[76] 2018
*Crypthecodinium cohnii* ATCC 30556	To comparative analyze *C. cohnii* mutants obtained from laboratory evolution	LC-MS and GC-MS	[15] 2018
*Scrippsiella trochoidea* CCMP 3099	To development single-cell metabolomics methodologies for small protists such as marine dinoflagellates	‘Single-probe’ MS	[163] 2018
*Ostreopsis* cf. *ovata*	To determine allelopathic interactions between the benthic toxic dinoflagellate *Ostreopsis* cf. *ovata* and a co-occurring diatom	LC-MS	[61] 2018

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
