# Peer review of "Omics Analysis for Dinoflagellates Biology Research"

_microorganisms, 2019, doi:10.3390/microorganisms7090288_

Round 1
Reviewer 1 Report
This is a very comprehensive and up-to-date description of progress in dinoflagellates using various 'omics methods. I enjoyed reading this article and learned a fair bit. I have only a few significant qualms that I suggest they address before publication:
Line 112-114, please replace "Symbiodinium" w/ the current taxonomic names, where appropriate. See this source. https://www.ncbi.nlm.nih.gov/pubmed/30100341
Line 120: It's really unlikely that the unidentified 2/3 of the Symbiodinium proteins originated through horizontal gene transfer. Even for prokaryotes, 2/3 is stretching the limits for HGT, and prokaryotes have machinery specifically for HGT (like plasmids, pili, etc), which eukaryotes don't. Most of these uncharacterized "dino-specific" genes are probably: (1) Fast-evolving sequences that have diverged too far from their eukaryotic orthologs to be recognized by sequence similarity; (2) Gene variants produced through gene fusion or domain-shuffling of pre-existing genes; or (3) Genes that evolved de-novo from mutations in noncoding sequences. Surely there are *some* dino-specific genes that have originated through HGT (including dinoviral nucleoproteins, histone-like proteins, type II RuBisCo, etc.) but this is rare.
Eukaryotes simply do not have the mechanisms to incorporate that many foreign genes, other than those involved in photosynthesis and respiration (which should still have orthologs in other eukaryotes). Don't get me wrong, especially in dinoflagellates, HGT is very interesting! But HGT should not be the default explanation for why genes can't be annotated.
Line 201: What are the advantages and disadvantages of these different methods? Are some used more often than others, and if so, why?
Line 263: "These results provide valuable insights into the molecular basis of thermal or light resistance of dinoflagellates and coral bleaching" Can you explain those insights? What have we learned, given that there was no transcriptomic response to thermal bleaching?
Line 133: Please define exometabolomics.
Since the authors are not native English speakers I also edited the manuscript for spelling and grammar. The edited pdf is available at the link below. I suggest they re-upload the draft after incorporating all of these changes.
https://drive.google.com/open?id=1dzLRkChM1iEKA71NikfWLVip-Sxxiw4J
I hope this feedback is useful to you and look forward to seeing more of your research in the future.
~Greg Gavelis
Author Response
Comments from review 1
1) This is a very comprehensive and up-to-date description of progress in dinoflagellates using various 'omics methods. I enjoyed reading this article and learned a fair bit. I have only a few significant qualms that I suggest they address before publication:
Reply: Thank very much for your appreciation of our work! We have made modifications according to your valuable suggestions. We hope these have addressed your concerns! Thanks again!
2) Line 112-114, please replace "Symbiodinium" w/ the current taxonomic names, where appropriate. See this source. https://www.ncbi.nlm.nih.gov/pubmed/30100341
Reply: Thanks for your suggestions! We are apologizing for using old taxonomic terms. We have replaced them with new taxonomic names in the revised manuscript (Lines 46-47; line 116; line 148; line 169; line 177; lines 180-181; line 187; line 278; line 284; line 318; line 323; line 423; line 426; line 433; line 505; line 509; Table 1; Table 2; Table 3). Thanks again!
3) Line 120: It's really unlikely that the unidentified 2/3 of the Symbiodinium proteins originated through horizontal gene transfer. Even for prokaryotes, 2/3 is stretching the limits for HGT, and prokaryotes have machinery specifically for HGT (like plasmids, pili, etc), which eukaryotes don't. Most of these uncharacterized "dino-specific" genes are probably: (1) Fast-evolving sequences that have diverged too far from their eukaryotic orthologs to be recognized by sequence similarity; (2) Gene variants produced through gene fusion or domain-shuffling of pre-existing genes; or (3) Genes that evolved de-novo from mutations in noncoding sequences. Surely there are *some* dino-specific genes that have originated through HGT (including dinoviral nucleoproteins, histone-like proteins, type II RuBisCo, etc.) but this is rare.
Eukaryotes simply do not have the mechanisms to incorporate that many foreign genes, other than those involved in photosynthesis and respiration (which should still have orthologs in other eukaryotes). Don't get me wrong, especially in dinoflagellates, HGT is very interesting! But HGT should not be the default explanation for why genes can't be annotated.
Reply: Thank you for your valuable suggestions! We totally agree with your opinion. It is too hasty to make such a conclusion that unidentified 2/3 of the Symbiodinaecea proteins are due to horizontal gene transfer. Therefore, we have deleted relevant sentences in the revised manuscript (Lines 122). Thanks again!
4) Line 201: What are the advantages and disadvantages of these different methods? Are some used more often than others, and if so, why?
Reply: Thanks for your comments! For microarrays approach, it is a high throughput and relatively inexpensive method, but is subject to a number of limitations, such as relying on existing genome sequence, high levels of background noise and a limited dynamic range of detection of different isoforms and allelic expression (Wang, et al., 2009; Royce, et al., 2007; Okoniewski, et al., 2006). For expressed sequence tag sequencing, it is based on Sanger sequencing with relatively high accuracy, but has many inherent disadvantages, such as low throughput, high cost and the lack of the ability of quantification (Wang et al., 2009; Boguski et al., 1994). For massively parallel signature sequencing (MPSS), it is a tag-based high throughput method and is still subject to the limitations (e.g. high cost, a limited dynamic range of detection of isoforms and disability of mapping a part of short tags to the reference genome) (Wang et al., 2009; Brenner et al., 2000). The last is RNA-Seq approach which shows clear advantages and has good application prospects in comparison with other methods. First, it is not limited to the detected transcripts that have corresponding genomic sequence, which attracts non-model organisms whose genome sequences have not yet been determined; second, it also has a high resolution to reveal the precise location of transcription boundaries, dynamic range to quantify gene expression level and ability to distinguish sequence variations, which are especially useful for complex transcriptomes (Wang et al., 2009). Finally, RNA-Seq approach has a relatively low background, high accuracy of quantitative expression level, high technological reproducibility and a much lower cost (Wang et al., 2009). Thus RNA-Seq technology is the most widely used approach during transcriptomic analysis of dinoflagellates in recent years.
We have added the information of comparing advantages and disadvantages of these different methods into the revised manuscript (Lines 216-235). We hope they have addressed your concerns! Thanks again!
5) Line 263: "These results provide valuable insights into the molecular basis of thermal or light resistance of dinoflagellates and coral bleaching" Can you explain those insights? What have we learned, given that there was no transcriptomic response to thermal bleaching?
Reply: Thanks for pointing it out. In the revised manuscript, we added an example of heat stress response in Fugacium kawagutii in the revised manuscript. It was discovered that 357 genes of F. kawaguti were differentially expressed under heat stress. In addition to heat shock and chaperonin protein, most of them were involved in regulating cell wall modulation and the transport of iron, oxygen, and major nutrients. These results suggested that cellular processes of cell wall modulation and ion/oxygen transport are likely to participate in thermal stress response in F. kawagutii (Lin et al., 2019).
For transcriptional changes of Symbiodiniaceae SSB01 in response to light, transcriptional levels of many genes seemed to be altered. For example, a cryptochrome 2 gene, regulators of Chromatin Condensation (RCC1), a light harvesting AcpPC protein and several cell adhesion proteins. The increased level of transcript of AcpPC gene suggested that it is involved in photo-protection and the dissipation of excess absorbed light energy. The decreased level of cell adhesion protein is related to the significant change of cell surface morphology, which may reflect the complexity of extracellular matrix (Xiang et al., 2015).
We have added the sentences to discuss what we have learned from transcriptional response of Symbiodiniaceae in response to thermal stress or light (Lines 279-283; lines 286-291) in the revised manuscript. Thanks again!
6) Line 133: Please define exometabolomics.
Reply: Thanks for your suggestions! Exometabolomics focuses on the research of small molecules that cells secrete into their environment (Silva et al., 2015). We have added it in the revised manuscript (Lines 442-443). Thanks again!
7) Since the authors are not native English speakers I also edited the manuscript for spelling and grammar. The edited pdf is available at the link below. I suggest they re-upload the draft after incorporating all of these changes.
https://drive.google.com/open?id=1dzLRkChM1iEKA71NikfWLVip-Sxxiw4J
Reply: Thank you very much for helping us to improve the quality of English writing! We appreciate it very much! We have incorporated all of these changes in the revised manuscript! Thanks again!
8) I hope this feedback is useful to you and look forward to seeing more of your research in the future.
Reply: Thanks again for your review and many valuable suggestions for improving our paper! Please let us know if you have any further questions, we will try out best to address them!
References
Boguski, M.S.; Tolstoshev, C.M.; Bassett, D.E. Gene discovery in dbEST. Science 1994, 265, 1993. Brenner, S.; Johnson, M.; Bridgham, J.; Golda, G.; Lloyd, D.H.;Johnson, D.e.a. Gene expression analysis by massively parallel signature sequencing (MPSS) on microbead arrays. Biotechnol. 2000, 18, 630-634. Lin S, Yu L, Zhang H. Transcriptomic Responses to Thermal Stress and Varied Phosphorus Conditions in Fugacium kawagutii. Microorganisms 2019, 7, 96. Okoniewski, M.J.; Miller, C.J. Hybridization interactions between probe sets in short oligo microarrays lead to spurious correlations. BMC Bioinformatics 2006, 7, 276-276. Royce, T.E.; Rozowsky, J.S.; Gerstein, M.B. Toward a universal microarray: prediction of gene expression through nearest-neighbor probe sequence identification. Nucleic Acids Res. 2007, 35, e99-e99. Silva, L.P., Northen, T.R. Exometabolomics and MSI: deconstructing how cells interact to transform their small molecule environment. Opin. Biotechnol. 2015, 34, 209-216. Wang, Z.; Gerstein, M.; Snyder, M. RNA-Seq: a revolutionary tool for transcriptomics. Rev. Genet. 2009, 10, 57-63. Xiang, T.; Nelson, W.; Rodriguez, J.; Tolleter, D.; Grossman, A.R. Symbiodinium transcriptome and global responses of cells to immediate changes in light intensity when grown under autotrophic or mixotrophic conditions. Plant J. 2015, 82, 67-80.
Reviewer 2 Report
The authors acknowledge the need for longer reads technologies to tackle the characteristically large genomes of dinoflagellates. They provide a comprehensive review of the current availability of dinoflagellate genomes and raise the point that these are generally focused on Symbiodiniaceae. The authors shed light on the potential for other omics (proteomics, and metabolomics) to further describe dinoflagellate biology, and touch on some of the existing methods used.
As an important and relatively well-studied dinoflagellate, Symbiodinaeceae is mentioned throughout the review, yet a fundamental publication – the taxonomic revision of Symbiodinium to the now widely accepted family Symbiodinaeceae – is not presented in this manuscript. The old taxonomic terms (e.g. clades) are used, and the species are incorrectly described. A lot of board statements about Symbiodiniaceae biology was based on observations in the F. kawagutti genome draft assembly, with no reference to the other 6 published genomes to support these observations. Some other seminal omics research on Symbiodiniaceae are also not included (especially in proteomics and metabolomics).
The authors provide one of the first reviews to touch on the status and potential for proteomics and metabolomics approaches to elucidate dinoflagellate functional biology. This is especially important given their importance in the health and functioning of marine ecosystems. The authors highlight some of the technologies available and how they have been applied. However, both the proteomics and metabolomics sections need expanding to include important missing omic technologies (e.g. LCMS and imaging mass specs such as MALDI-MS), and some important manuscripts are missing from their references. While the authors highlight the technological challenges facing proteomic and metabolomic studies, one important way to forward proteomics and metabolomics discovery would be to describe the proteome/metabolome of a dinoflagellate – having such a reference proteome/metabolome would greatly increase the rate of discovery in dinoflagellate functional biology but is overlooked in this review. This could also reveal biomarkers for certain functional characteristics or responses of dinoflagellates.
As acknowledged (albeit briefly) by the authors, some great strides in dinoflagellate functional biology could be made by integrating the omics. While extensively (9 years ago) reviewed for microbial biology elsewhere (Zhang et al 2010), this could be expanded on in a separate paragraph, including some more recent examples in which this has been successfully applied e.g. to compare Symbiodinaceae performance when in symbiosis (Matthews et al 2017).
Overall, there is a need for the systematic review of omics analysis to forward our understanding of dinoflagellate biology. This review will greatly benefit from some revisions in the language to provide clarification, as the English used is confusing (and incorrect) at times.
Line 41: example (and reference) for dinoflagellates being a key component of human food
Line 45: Symbiodinium has now been reclassified as a family (Symbiodiniaceae) – see LaJeunesse et al 2018 https://doi.org/10.1016/j.cub.2018.07.008
Line 48: Symbiodiniaceae provide much more than energy metabolites to corals. They translocate antioxidants, signalling compounds, and are vitally important in nutrient cycling.
Line 57: a sharp increase in dinoflagellates and bacteria etc.
Line 68 and 70: a repeat of “In addition”.
Line 92: “symbiosis with coral reef” makes no sense. Do you mean corals? If so, there are other organisms, such as clams and foraminifera (so not just corals) that dinoflagellates associate with.
Lines 98 -100: This sentence is unclear. Also, the high cost is not the only reasons for dinoflagellate nuclear genomes being “out of reach” - their large sizes make sequencing the entire genome tricky.
Line 112-114: these species names are out of date.
Line 115: which Symbiodiniaceae species? Need to rephrase as I think you mean Symbiodiniaceae in general?
Line 126: The point of this sentence is unclear.
Line 157: Cite Sproles et al. 2018 (https://doi.org/10.1016/j.ympev.2017.12.007)
Line 160-161: insert references for the statement on Mycosporine-like amino acids.
Line 165: what is K. kawagutti? Do you mean F Kawagutti? F. kawagutti is a member of clade F.
Line 304-308: another important tool for proteomic analyses is LCMS analysis (for example, see Sproles et al 2019: https://doi.org/10.1038/s41396-019-0437-5
Line 318: Besides, environmental stress, proteomics has also been used to compare the performance of different Symbiodiniaceae species: see Medrano et al 2019 https://doi.org/10.3389/fmicb.2019.01153and Sproles et al 2019: https://doi.org/10.1038/s41396-019-0437-5
Line 343: I wouldn’t necessarily say the technologies are immature; there are a number of incredibly high-resolution technologies able to detect and identify changes in metabolite profiles – however, our understanding of how the changes affect the overall biological function is still in its infancy. For example – is a higher metabolite abundance indicative of action expression and turnover, or inactive pathways and accumulation? It’s difficult to know simply from metabolite profiles, and this is why integrating the omics has a great deal of power to determine the overarching biological pathways activated under specific conditions.
Line 347: missing reference: Matthews et al 2018: https://doi.org/10.1098/rspb.2018.2336
Line 361: 13C flux has also been used to compare metabolite exchange by different Symbiodiniaceae species (Matthews et al 2018: https://doi.org/10.1098/rspb.2018.2336)
Line 373: I would say metabolomics is “closer” to the phenotype, rather than “better”
Line 376: do you mean easier than proteomics?
Author Response
Comments from review 2
1) The authors acknowledge the need for longer reads technologies to tackle the characteristically large genomes of dinoflagellates. They provide a comprehensive review of the current availability of dinoflagellate genomes and raise the point that these are generally focused on Symbiodiniaceae. The authors shed light on the potential for other omics (proteomics, and metabolomics) to further describe dinoflagellate biology, and touch on some of the existing methods used.
Reply: Thanks very much for the accurate summary! We have revised the manuscript according to all valuable suggestions from you and another anonymous reviewer, and the new details related to comments have also been added into the revised manuscript. Thanks again!
2) As an important and relatively well-studied dinoflagellate, Symbiodiniaceae is mentioned throughout the review, yet a fundamental publication – the taxonomic revision of Symbiodinium to the now widely accepted family Symbiodiniaceae – is not presented in this manuscript. The old taxonomic terms (e.g. clades) are used, and the species are incorrectly described. A lot of board statements about Symbiodiniaceae biology was based on observations in the F. kawagutti genome draft assembly, with no reference to the other 6 published genomes to support these observations. Some other seminal omics research on Symbiodiniaceae are also not included (especially in proteomics and metabolomics).
Reply: Thanks very much for your suggestions! We are apologizing for using old taxonomic terms. We have replaced them with new taxonomic names throughout the revised manuscript. In addition to Fugacium kawagutti genome draft, we have added the other six published genomes to explain the characteristics of Symbiodiniaceae biology, genetic differences between Symbiodiniaceae species and the molecular basis of coral-Symbiodiniaceae symbiosis (Lines 149-184). Other seminal proteomics and metabolomics research on Symbiodiniaceae are also included in the revised manuscript (Lines 373-378; lines 382-391; lines 426-431; line 434-438). For example, proteomics has been employed to discovery proteins of Symbiodiniaceae (Peng et al., 2008) and to reveal proteins involved in symbiosis and their responses to environmental stress ( Weston et al., 2012) (Lines 373-378). Metabolomics was used to explore the reason why acidification markedly inhibited the growth of B. minutum (Jiang et al., 2019) (Lines 426-431). Thanks again!
3) The authors provide one of the first reviews to touch on the status and potential for proteomics and metabolomics approaches to elucidate dinoflagellate functional biology. This is especially important given their importance in the health and functioning of marine ecosystems. The authors highlight some of the technologies available and how they have been applied. However, both the proteomics and metabolomics sections need expanding to include important missing omic technologies (e.g. LCMS and imaging mass specs such as MALDI-MS), and some important manuscripts are missing from their references.
Reply: Thanks for your valuable suggestions! We are apologizing for missing some information in the original manuscripts. We have added them in the revised manuscripts (reference 125, 126, 132 and 141). Meanwhile, we have expanded the proteomics and metabolomics sections to include important omic technologies (Lines 333-364, 375-378, 382-389, 426-431, 434-438). Thanks again!
4) While the authors highlight the technological challenges facing proteomic and metabolomic studies, one important way to forward proteomics and metabolomics discovery would be to describe the proteome/metabolome of a dinoflagellate – having such a reference proteome/metabolome would greatly increase the rate of discovery in dinoflagellate functional biology but is overlooked in this review. This could also reveal biomarkers for certain functional characteristics or responses of dinoflagellates.
Reply: Thanks for pointing it out! We totally agree with your comments. We added sentences to highlight the importance of strengthening the description of the complete proteome/metabolome of a dinoflagellate, and establishing reference proteome/metabolome of a dinoflagellate in the "Conclusions and prospective" sections of the revised manuscript (Lines 494-497). Thanks again!
5) As acknowledged (albeit briefly) by the authors, some great strides in dinoflagellate functional biology could be made by integrating the omics. While extensively (9 years ago) reviewed for microbial biology elsewhere (Zhang et al 2010), this could be expanded on in a separate paragraph, including some more recent examples in which this has been successfully applied e.g. to compare Symbiodinaceae performance when in symbiosis (Matthews et al 2017).
Reply: Thank you for your valuable suggestions! We appreciate it very much! The "multi-omics" method has been proved to be a powerful tool for understanding the functional principles and dynamics of total cellular systems (Zhang et al., 2010; Mootha et al., 2003a; Mootha et al., 2003b). For dinoflagellate, "multi-omics" approach has been demonstrated to be powerful in dinoflagellate functional biology research. We have added a separate paragraph to explain it (Lines 455-483). It includes: i) characterization of the molecular and physiological processes of DHA synthesis in fed-batch fermentation of Crypthecodinium cohnii (Pei et al., 2017), ii) comparing Symbiodinaceae performance in different host-symbiont pairings (Matthews et al 2017), iii) temperature adaptation mechanism of Zooxanthellate cnidarians (Cziesielski et al., 2018). Thanks again!
6) Overall, there is a need for the systematic review of omics analysis to forward our understanding of dinoflagellate biology. This review will greatly benefit from some revisions in the language to provide clarification, as the English used is confusing (and incorrect) at times.
Reply: Thanks for your valuable advices! We have carefully proofread the manuscript several times and have tried our best to identify and correct issues related to English use. Thanks again!
7) Line 41: example (and reference) for dinoflagellates being a key component of human food
Reply: Thanks for your comments! DHA which has various beneficial effects to human health, such as improving cognitive development in infants and inhibiting hypertension, inflammation and certain cancers (Akbar et al, 2005; Calon et al., 2004; Hong et al., 2003), cannot be synthesized by human being, but accumulates in heterotrophic dinoflagellates Crypthecodinium cohnii (Mendes et al., 2008). We have added it in the revised manuscript (Lines 40-41; Lines 50-53). Thanks again!
8) Line 45: Symbiodinium has now been reclassified as a family (Symbiodiniaceae) – see LaJeunesse et al 2018 https://doi.org/10.1016/j.cub.2018.07.008
Reply: Thanks very much! We have revised it throughout the revised manuscript.
9)Line 48: Symbiodiniaceae provide much more than energy metabolites to corals. They translocate antioxidants, signalling compounds, and are vitally important in nutrient cycling.
Reply: Thanks for the suggestions! We have revised it (Lines 47-49).
10)Line 57: a sharp increase in dinoflagellates and bacteria etc.
Reply: Thank you! We have revised it (Lines 57-58).
11) Line 68 and 70: a repeat of “In addition”.
Reply: Thank you for pointing it out! We have deleted the second one. Thanks again!
12) Line 92: “symbiosis with coral reef” makes no sense. Do you mean corals? If so, there are other organisms, such as clams and foraminifera (so not just corals) that dinoflagellates associate with.
Reply: Thank you for pointing it out! We have revised "symbiosis with coral reef" into "symbiosis" in the revised manuscript (Line 94).
13)Lines 98 -100: This sentence is unclear. Also, the high cost is not the only reasons for dinoflagellate nuclear genomes being “out of reach” - their large sizes make sequencing the entire genome tricky.
Reply: Thanks for your valuable suggestions! We have revised it (Lines 100-102).
14) Line 112-114: these species names are out of date.
Reply: We are apologizing for using old taxonomic terms. We have revised it (Lines 114-116).
15)Line 115: which Symbiodiniaceae species? Need to rephrase as I think you mean Symbiodiniaceae in general?
Reply: Thank you for the comment! It referred to Breviolim minutum and Fugacium kawagutii. (Line 117). Thanks again!
16)Line 126: The point of this sentence is unclear.
Reply: Thank you for your suggestions! We have revised it into "What’s more, a novel promoter element (motifs TTTT instead of TATA box used by other eukaryotes) in F. kawagutii genome, and a microRNA system potentially regulating gene expression in both symbiont and coral were observed" (lines 126-127).
17)Line 157: Cite Sproles et al. 2018 (https://doi.org/10.1016/j.ympev.2017.12.007)
Reply: Thanks! We have included it in the revised manuscript (reference 84) (line 160).
18) Line 160-161: insert references for the statement on Mycosporine-like amino acids.
Reply: Thank you for your suggestions! We have inserted references for the statement on Mycosporine-like amino acids (reference 28 and 85) (lines 164-167).
19) Line 165: what is K. kawagutti? Do you mean F Kawagutti? F. kawagutti is a member of clade F.
Reply: Thanks for the comment! We apologize for using old taxonomic terms of Symbiodiniaceae. We have revised it in the revised manuscript (Lines 171-172).
20) Line 304-308: another important tool for proteomic analyses is LCMS analysis (for example, see Sproles et al 2019: https://doi.org/10.1038/s41396-019-0437-5
Reply: Thanks for your suggestions! LCMS analysis is an important tool for proteomic analyses (Sproles et al., 2019), and it also belongs to gel-free profiling procedures which rely on multidimensional separations coupling micro-scale separations with automated tandem mass spectrometry (Geert et al., 2005; Nie et al., 2008). We have revised the relevant sentences (Lines 330-340; lines 373-378) and included the references (reference 126,132) in the revised manuscript.
21) Line 318: Besides, environmental stress, proteomics has also been used to compare the performance of different Symbiodiniaceae species: see Medrano et al 2019 https://doi.org/10.3389/fmicb.2019.01153and Sproles et al 2019: https://doi.org/10.1038/s41396-019-0437-5
Reply: Thanks for pointing it out! We have cited the two references (reference 126 and 134) added sentences to compare the performance of different Symbiodiniaceae species in different host-symbiont pairings (Lines 382-391). Thanks again!
22) Line 343: I wouldn’t necessarily say the technologies are immature; there are a number of incredibly high-resolution technologies able to detect and identify changes in metabolite profiles – however, our understanding of how the changes affect the overall biological function is still in its infancy. For example – is a higher metabolite abundance indicative of action expression and turnover, or inactive pathways and accumulation? It’s difficult to know simply from metabolite profiles, and this is why integrating the omics has a great deal of power to determine the overarching biological pathways activated under specific conditions.
Reply: Thanks for your valuable suggestions! We totally agree with your comment! We have deleted "technologies are immature" and revised it into "our understanding of how these differently expressed metabolites affect the overall biological function is still in infancy" (Lines 409-411).
23) Line 347: missing reference: Matthews et al 2018: https://doi.org/10.1098/rspb.2018.2336
Reply: Thanks for your suggestions! We have added it (reference 141) in the revised manuscript (line 414).
24) Line 361: 13C flux has also been used to compare metabolite exchange by different Symbiodiniaceae species
(Matthews et al 2018: https://doi.org/10.1098/rspb.2018.2336)
Reply: Thanks for your valuable suggestions! We have included it (reference 141) and added sentences to discuss it in the revised manuscript (Lines 434-438) in the revised manuscript.
25) Line 373: I would say metabolomics is “closer” to the phenotype, rather than “better”
Reply: Thanks for your suggestion! We have used "closer" to replace "better" (Line 451) in the revised manuscript.
26) Line 376: do you mean easier than proteomics?
Reply: Thanks for the comment! We are apologizing for the misunderstanding. Actually, it is hard to compare the difficulty of methods between proteomics and metabolomics. We have deleted it in the revised manuscript (Lines 452-454).
References
Akbar, M.; Calderon, F.; Wen, Z.; Kim, H.Y. Docosahexaenoic acid: A positive modulator of Akt signaling in neuronal survival. P NATL ACAD SCI USA 2005, 102, 10858-10863. Calon, F.; Lim, G.P.; Yang, F.; Morihara, T.; Teter, B.; Ubeda, O.; Rostaing, P.; Triller, A.; Jr, S.N.; Ashe, K.H. Docosahexaenoic acid protects from dendritic pathology in an Alzheimer's disease mouse model. Neuron 2004, 43, 633-645. Cziesielski, M.J.; Liew, Y.J.; Cui, G.; Schmidt-Roach, S.; Campana, S.; Marondedze, C.; Aranda, M. Multi-omics analysis of thermal stress response in a zooxanthellate cnidarian reveals the importance of associating with thermotolerant symbionts. Biol. Sci. 2018, 285, 1877. Geert, B.; Evy, V.; Arnold, D.l.; Liliane, S. Gel-based versus gel-free proteomics: a review. Chem. High. T. Scr. 2005, 8, 669-677. Hong, S.; Gronert, K.; Devchand, P.R.; Moussignac, R.L.; Serhan, C.N. Novel docosatrienes and 17s-resolvins generated from docosahexaenoic acid in murine brain, human blood, and glial cells autacoids in anti-inflammation. Biol. Chem. 2003, 278, 14677-14687. Jiang, J.; Lu, Y. Metabolite profiling of Breviolum minutum in response to acidification. Toxicol. 2019, 213, 105215. Matthews, J.L.; Crowder, C.M.; Oakley, C.A.; Lutz, A.; Roessner, U.; Meyer, E.; Grossman, A.R.; Weis, V.M.; Davy, S.K. Optimal nutrient exchange and immune responses operate in partner specificity in the cnidarian-dinoflagellate symbiosis. NATL. ACAD. SCI. USA. 2017, 114, 13194-13199. Mendes, A.; Reis, A.; Vasconcelos, R.; Guerra, P.; Silva, T.L.D. Crypthecodinium cohnii with emphasis on DHA production: a review. Appl. Phycol. 2008, 21, 199-214. Mootha, V.K.; Jakob, B.; Olsen, J.V.; Majbrit, H.; Wisniewski, J.R.; Erich, S.; Bolouri, M.S.; Ray, H.N.; Smita, S.; Michael, K. Integrated analysis of protein composition, tissue diversity, and gene regulation in mouse mitochondria. Cell 2003a, 115, 629-640. Mootha, V.K.; Pierre, L.; Kathleen, M.; Jakob, B.; Michael, R.; Majbrit, H.; Terrye, D.; Amelie, V.; Robert, S.; Fenghao, X. Identification of a gene causing human cytochrome c oxidase deficiency by integrative genomics. NATL. ACAD. SCI. USA. 2003b, 100, 605-610. Nie, L.; Wu, G.; Zhang, W. Statistical application and challenges in global gel-free proteomic analysis by mass spectrometry. Rev. Biotechnol 2008, 28, 297. Pei, G.S.; Li, X.R.; Liu, L.S.; Liu, J.; Wang, F.Z.; Chen, L.; Zhang, W.W. De novo transcriptomic and metabolomic analysis of docosahexaenoic acid (DHA)-producing Crypthecodinium cohnii during fed-batch fermentation. Algal Res. 2017, 26, 380-391. Peng, S.E.; Luo, Y.J.; Huang, H.J.; Lee, I.T.; Hou, L.S.; Chen, W.N.U.; Fang, L.S.; Chen, C.S. Isolation of tissue layers in hermatypic corals by N -acetylcysteine: morphological and proteomic examinations. Coral Reefs 2008, 27, 133-142. Sproles, A.E.; Oakley, C.A.; Matthews, J.L.; Peng, L.; Owen, J.G.; Grossman, A.R.; Weis, V.M.; Davy, S.K. Proteomics quantifies protein expression changes in a model cnidarian colonised by a thermally tolerant but suboptimal symbiont. The ISME J. 2019, 10.1038/s41396-019-0437-5. Weston, A.J.; Dunlap, W.C.; Shick, J.M.; Klueter, A.; Iglic, K.; Vukelic, A.; Starcevic, A.; Ward, M.; Wells, M.L.; Trick, C.G.; Long, P.F. A profile of an endosymbiont-enriched fraction of the coral Stylophora pistillata reveals proteins relevant to microbial-host interactions. Cell. Proteomics 2012, 11, M111 015487. Zhang, W.; Li, F.;Nie, L. Integrating multiple 'omics' analysis for microbial biology: application and methodologies. Microbiology 2010, 156, 287-301.
Round 2
Reviewer 2 Report
I appreciate the authors' careful review and incorporation of my comments. The manuscript is greatly improved as a result, especially in the quality of the english language. That said, there are still a few grammatical errors and the manuscript could still benefit from some further english editing. For example (but not limited, to so the authors should continue to review the english beyond these suggestions):
Line 52 to 5: ...human beings, but accumulates in heterotrophic dinoflagellates...
Line 58: delete etc
Line 100: change "referred to" to "such as" or "referred to as"
Other than that, I congratulate the authors to a great piece of work and believe that this review is acceptable for publication once these comments have been addressed.
Author Response
Comments from review 2
1) I appreciate the authors' careful review and incorporation of my comments. The manuscript is greatly improved as a result, especially in the quality of the English language. That said, there are still a few grammatical errors and the manuscript could still benefit from some further English editing. For example (but not limited, to so the authors should continue to review the English beyond these suggestions):
Reply: Thanks for your appreciation of our work! We have incorporated all of these changes in the revised manuscript and carefully proofread the manuscript several times and have tried our best to identify and correct issues related to English use. Thanks again!
2) Line 52 to 5: ...human beings, but accumulates in heterotrophic dinoflagellates...
Reply: Thank you for pointing it out! We have revised it (line 52). Thanks again!
3) Line 58: delete etc
Reply: Thank you for pointing it out! We have deleted “etc” in the revised manuscript (line 58). Thanks again!
4) Line 100: change "referred to" to "such as" or "referred to as"
Reply: Thank you for pointing it out! We have revised it (line 100). Thanks again!
Other than that, I congratulate the authors to a great piece of work and believe that this review is acceptable for publication once these comments have been addressed.
Reply: Thanks again for your review and many valuable suggestions for improving our paper!
